# Predicting the Appearance of Hypotension during Hemodialysis Sessions Using Machine Learning Classifiers

**DOI:** 10.3390/ijerph18052364

**Published:** 2021-02-28

**Authors:** Juan A. Gómez-Pulido, José M. Gómez-Pulido, Diego Rodríguez-Puyol, María-Luz Polo-Luque, Miguel Vargas-Lombardo

**Affiliations:** 1Department of Technologies of Computers and Communications, University of Extremadura, 10003 Cáceres, Spain; jangomez@unex.es; 2Ramón y Cajal Institute for Health Research (IRYCIS), Department of Computer Science, University of Alcalá, 28801 Alcalá de Henares, Spain; 3Department of Medicine and Medical Specialties, Research Foundation of the University Hospital Príncipe de Asturias, IRYCIS, Alcalá University, 28801 Alcalá de Henares, Madrid, Spain; 4Department of Nursing and Physiotherapy, University of Alcalá, 28801 Alcalá de Henares, Spain; mariluz.polo@uah.es; 5E-Health and Supercomputing Research Group (GISES), Technological University of Panama, Panama City 0819-07289, Panama; miguel.vargas@utp.ac.pa

**Keywords:** hemodialysis, hypotension, clinical monitoring, supervised learning, decision trees, support vector machines

## Abstract

A patient suffering from advanced chronic renal disease undergoes several dialysis sessions on different dates. Several clinical parameters are monitored during the different hours of any of these sessions. These parameters, together with the information provided by other parameters of analytical nature, can be very useful to determine the probability that a patient may suffer from hypotension during the session, which should be specially watched since it represents a proven factor of possible mortality. However, the analytical information is not always available to the healthcare personnel, or it is far in time, so the clinical parameters monitored during the session become key to the prevention of hypotension. This article presents an investigation to predict the appearance of hypotension during a dialysis session, using predictive models trained from a large dialysis database, which contains the clinical information of 98,015 sessions corresponding to 758 patients. The prediction model takes into account up to 22 clinical parameters measured five times during the session, as well as the gender and age of the patient. This model was trained by means of machine learning classifiers, providing a success in the prediction higher than 80%.

## 1. Background and Objectives

### 1.1. Clinical Consequences of Hemodynamic Changes in Hemodialysis Patients

Dialysis and transplantation have allowed many patients with advanced chronic renal disease (ERC) to remain alive and have a good quality of life for many years. Although the desirable treatment in these patients is early kidney graft, this is not always possible in part by the limited availability of organs and partly by the characteristics of the patients. Consequently, in 2014, there were 26,533 patients in Spain who needed dialytic techniques to stay alive, and of these 23,519 patients were subsidiaries of a technique known as hemodialysis (HD). If we compare these digits with the prevalence of other diseases, they are not very significant. However, the impact on individuals and families is brutal, and the economic cost is disproportionate as 2.5% of all Spanish health spending is devoted to the treatment of patients with terminal ERC [1].

In hemodialysis, the internal environment of patients is purged periodically using artificial filters, called dialyzers, through a system that connects the blood of individuals with special fluids, designed to reproduce the normal composition of body fluids, through a semipermeable membrane [2]. This is a very sophisticated procedure, controlled by specially designed devices, generically known as dialysis monitors that control and adjust the process. For clinical reasons, and because of the quality of life of patients, hemodialysis is usually organized in sessions lasting an average of 4 h and carried out about three times a week. In these sessions, the patient blood is intensively purified, allowing a reasonable replacement of renal function, consistent with good expectations and quality of life.

The process of intermittent blood purification, with approximate interdialytic intervals of 48 h, constitutes a controlled aggression for patients. In fact, the potential complications during dialysis and interdialytic periods constitute one of the most important areas of work for nephrologists and dialysis nurses. Among all these complications, the one that has received most attention is what we could call hemodynamic changes. As hemodialysis patients do not urinate or urinate little, fluids accumulate at the cardiovascular level in the interdialytic period, with the subsequent overload in the vascular tree and hearth. Hypertension and a marked tendency to cardiac insufficiency are prominent characteristics of hemodialysis patients [3,4]. On the other hand, the dialytic treatment session has to ensure the removal of fluids accumulated in the interdialytic intervals, thus conditioning the removal of significant amounts of fluids, between 1 and 6 L, in relatively short periods of time. In this context, patients may exhibit acute decreases of blood pressure, with the subsequent decrease of cerebral, heart, or other organ perfusion, developing clinical symptoms or even very serious complications. Moreover, this rapid loss of fluids can make the patient feel unwell in variable periods of time after finishing dialysis, which greatly diminishes the quality of life of patients [5,6].

In the early days of hemodialysis, intradialytic hypotension was one of the most important clinical problems of the procedure. The gradual improvement of dialyzers and dialysis fluids and monitors, as well as the intensive research on the mechanisms responsible for the problem, has conditioned a progressively better tolerance to the treatment, with less dramatic blood pressure reductions and better prevention and treatment of these complications. One reason for the relative success of all these measures is the existence of a comprehensive and extensive system of registration and control of all intradialytic variables, which can be modified for improving hemodynamic tolerance. These extensive records include monitor, dialyzer, and patient variables. Nursing plays a key role in all these records, but increasingly monitors control, virtually “on-line”, changes in many of these parameters.

Despite all the advances mentioned above, the frequency of hypotension during dialysis remains around 15% of the sessions, which is a very significant prevalence [7]. The nursing staff attending these patients has assumed that it is almost inevitable and, although active in preventing the problem by trying to control diet and drug therapy and maximally optimizing dialysis, they must manage everyday a significant number of hypotensions.

On the other hand, fluid overload remains a major clinical problem in these patients. Hypertension is easily and regularly detected, and the wide spectra of potent drugs available to decrease blood pressure have contributed to a better management of this problem. Nevertheless, cardiovascular events remain responsible for about 50% of deaths of hemodialysis patients, and the cardiac dysfunction conditioned by inadequate fluid management is one of the most relevant threats for them [8].

In addition, we must consider the need for renal replacement therapy has increased alarmingly in recent years in Europe. This is mainly due to the progressive aging of the population, together with the higher diabetes and cardiovascular disease prevalence. Data extracted from the Spanish Registry of Renal Patients confirm this trend, with some regional differences to be taken into account.

In one year, a standard hemodialysis patient receives 156 hemodialysis sessions. In each session, at least 50 variables are recorded. That means that each patient has about 7800 records/year. Additionally, clinical and analytical parameters are regularly recorded in these patients, and probably more than 50 data each 15 days are registered. All these data are currently being stored in specifically designed applications, together with the demographic and clinical characteristics of the patients. If we consider the “Principe de Asturias” Hospital at Madrid, Spain, with a mean of 60 patients on hemodialysis, we can store about 540,000 data/year. Analyzing the patterns of variability, and their relationship with the patient’s problems in a quantitatively important number of data, by using techniques of big data and data mining, constitutes an innovative approach in healthcare field, and we can reduce the prevalence of intradialytic hypotension and cardiac insufficiency beyond the limits that we consider “normal” at present.

### 1.2. Evidence Supporting the Use of Massive Data Analysis in Healthcare

The application of big data in health will become intensive both for the prevention of diseases and for the rest of health activities: research, diagnosis, treatment, and evaluation of therapeutic results. Health research is based on the concept of medical evidence and clinical trials are always accredited by the evidence. With analytical tools, big data will cross the information of all registered patients to obtain the most complete research study that a health professional can imagine. From their knowledge and experience, doctors use all possible information to diagnose correctly. The data to take into account are many: medical history, pathologies, diagnostic tests, health habits, adherence to treatments, etc.

Having large databases of clinical evidence, doctors can contrast their personalized prescription with the recommendation “prescribed” by big data technologies [9], which would include the best decisions of expert colleagues. In this way, the health professional can verify which treatment provides the best result in patients with the same profile. To close the cycle of health activities, the doctor will follow up on the patient and record the effectiveness of the treatment. Thus, each particular case will be new evidence about said disease. The application of big data in the prevention of diseases will be the ideal tool to predict both collectively and individually, the likelihood of suffering the next disease that lurks after the preventive analysis of all our information. At a more personalized level, it will be possible to determine the persons whose profile is at risk of suffering from certain diseases in relation to their usual activity, their consumption profile, their genomics, etc. There are many recent works supporting this approach, particularly by applying machine learning techniques, such as predicting heart diseases [10], stays in intensive care units [11], and cervical cancer detection [12], among many others.

Currently, in the field of nephrology, intradialytic hypotensions continue to be one of the main dialysis unit clinical problems. They keep conditioning most of the therapeutic interventions of nurses, constitute a vital risk for patients, and reduce their quality of life. The mechanisms responsible for hypotensions have been studied repeatedly, establishing factors clearly related to their appearance, such as the composition of hemodialysis bath, its temperature, or time sequence of circulating blood volumes variations. In the latest years, in addition to all the accomplished clinical works by dialysis unit practitioners, the development of the biofeedback systems is contributing to control this problem, but hypotensions persist. For those reasons, we intend to detect a combination of clinical and analytical parameters associated with the appearance of hypotensions using massive data analysis. In this way, if some of these parameters could be adjusted, the incidence of this problem could be reduced. To do this, we intend to address this issue with techniques of machine learning, fundamentally because its applicability is perfectly possible.

#### 1.2.1. Present Status of Massive Data Analysis and Pattern Recognition in Health

Big data refers to huge data, and so large and complex information processing makes it very difficult management tools using conventional databases. The main problem is how to access, distribute, and use this vast amount of “unstructured” data that remains unused because of the difficulty of treating them effectively [13]. Therefore, it represents an opportunity for innovators and all those who care about health, since it allows substantially increase the possibility of more effective information data and lower mortality rates of patients.

Among experts and professionals, there is a high consensus on the theoretical benefits of the application of big data to the world of health and, in general, of all aspects of digital health, configuring itself as the great hope to maintain the quality of healthcare in Europe—a continent, with in an increasing number of elderly people, and therefore, with a high prevalence of chronic pathologies, that results in an enormous health expenditure associated with it.

Currently, most of the work related to the application of big data in health refers either to technical aspects of computer or computational type or to the benefits that can be expected from its application, from the needs and barriers that must be overcome, or future trends. However, it is already clear that this technology is not fully established and still has a long way to go. There is scant real deployment of big data solutions in the health field, considering that the health sector is far from the progress achieved in other areas, such as the financial sector or the large technology sector.

The vast majority of real and concrete big data experiences in the field of health are manifested only in the form of pilot projects, which do not usually crystallize into long-term initiatives. The main reasons for the non-consolidation of pilots are argued as: lack of strategic vision in the Spanish health sector as a whole; the lack of clear incentives to continue with the implementation of these projects; and issues related to a certain fear in the health sector to take false steps when carrying out definitive actions for its implementation. In some cases, even the issue of data protection regulations is adduced as justification for their non-implementation.

For these reasons, we propose to give real solutions to some of the main clinical problems that often occur in intradialysis. Our approach consists of processing the information using machine learning techniques. Hence, it is necessary to undertake challenges such as the analyzing, capturing, collecting, searching, sharing, storing, transferring, visualizing, etc., a large volume of information to obtain on-line knowledge, guaranteeing always the protection of personal data.

#### 1.2.2. Previous Experiences of Massive Data Analysis and Pattern Recognition in Healthcare

Health area generates a large amount of data, useful for improving the health care system. Usually, this information is stored in Electronic Health Record systems (EHRs) following different formats and processes. Data mining is one of the most valuable ways to manage the heterogeneous sources to extract relevant information, particularly for finding hidden medical data patterns, useful for clinical diagnosis [14,15,16]. Furthermore, data mining helps to reach the goals of diagnosis and treatment in healthcare, pursuing the healthcare quality as Health Care Output (HCO).

Health informatics is a combination of information science and computer science within the realm of healthcare. There are many current areas of research within the field of health informatics, including bioinformatics, image informatics, clinical informatics, public health informatics, and translational bio-informatics. Research done in health informatics includes data acquisition, retrieval, storage and analytics [17] employing data mining techniques and mining web [18].

On the other hand, clinical informatics research involves making predictions that can help physicians make better, faster, and more accurate decisions about their patients through analysis of patient data. Clinical questions are the most important question level in health informatics as it works directly with the patient. This is where a confusion can arise with the term “clinical” when found in research, as all health informatics research is performed with the eventual goal of predicting “clinical” events (directly or indirectly). This confusion is the reason for defining clinical informatics as only research that directly uses patient data. Next, we analyze some research performed through data mining.

In recent years, several projects related to big data have been developed in the field of digital health around the world. The results obtained in these projects lead us to conclude that, in the future, it is possible to reduce health costs through the efficient use of big data and appropriate tools. Medical examination, diagnosis, and prescriptions, most of the activities of the doctor, will be performed with computers, better managing data. Computers could replace large amounts of managerial medical work, broadening its powers to more useful ones.

#### 1.2.3. Techniques Massive Data Analysis and Its Applicability to the Field of Hemodialysis

The health model and the general health sector is one of the sectors where big data has greater impact today and where its applications grow in spectacular fashion for both the medical area, as also for the areas of data analysis (medical records, clinical analysis, etc.), management of health centres, hospital management, and scientific documentation (generation, storage, operation, etc.). The potential for big data in medicine lies in the possibility of combining the traditional data with new forms of data at both individual and population levels, i.e., integrate structured and unstructured data. Indeed, in the health sector, a huge quantities and varieties of structured, semi-structured, and unstructured data are generated. Structured data are data that can be stored, accessed, analyzed, and manipulated by machines, usually in data table mode. Unstructured data are quite the opposite. Structured data are the classic patient data (name, age, sex, etc.) and unstructured data are paper prescriptions, medical records, handwritten notes of doctors and nurses, voice recordings, X-rays, scans, MRI, TAC, and other medical imaging.

Technological advances are creating a new avalanche of data of all kinds that come from the most varied devices, sensors, medical devices, hospital data, as well as data from social media (social networks, blogs, wikis, podcasts, etc.), smartphones. Therefore, with big data, major savings and improvements in the health sector can be produced, and specifically in the work that is done in hemodialysis.

It is recommended address three emerging trends in the use of data that are of great importance: (1) working with limited datasets; (2) combining a variety of data; and (3) grouping data to improve results. Thus, medical research can improve greatly, being able to absorb a huge amount of data (monitoring, histories, treatments, etc.), especially unstructured data, and organize or structure them to define the causes of disease and establish better solutions.

Therefore, big data is extremely useful to predict, prevent, and customize treatment of diseases. It can be applied practically to almost all sectors of health, but in particular we can already mention some in which are found the greatest challenges: (1) genomic research and genome sequencing; (2) clinical operations; (3) self-help and cooperation of citizens; (4) personalized medicine for all; (5) improved personalized patient care; (6) remote monitoring of patients; (7) monitoring chronic patients; and (7) improvements in medical processes.

The huge amount of information available requires tools to be monitored, processed, screened, and exploited for the benefit of the patient, health professional knowledge, and in training future doctors and nurses. Another important feature is the increased use every day to monitor patient sensors. The sensors of all kinds that are part of the Internet of Things facilitate the transmission and reception of data from patients in hospitals, helping both face medical care and the patient’s home. The Internet of Things is one of the pillars of big data and allows us to accumulate more data from patients each day, thereby improving the previous diagnosis, through the comparative analysis of profiles with the same diagnosis.

Massive data analysis is combined with data mining and machine learning techniques for direct application in any medical field. There are numerous works where these techniques have been applied in the field of dialysis [19,20,21,22,23,24].

### 1.3. Hypothesis and Objectives

It is possible to detect a combination of clinical parameters that are associated with hypotension appearance or cardiac insufficiency development. If any combination of these parameters is measured for a patient during a dialysis session, the possible appearance of hypotension could be predicted, therefore reducing the incidence of this problem.

The objectives of the research were: (1) extract and build datasets from a large hemodialysis database, able to be handled by massive data processing tools; and (2) predict the appearance of hypotension from existing clinical parameter patterns.

### 1.4. Hemodialysis Database

Our research starts from a large database containing the clinical information of 758 patients who underwent hemodialysis sessions. These data were collected at Hospital Príncipe de Asturias, Madrid, Spain, during a period of almost four years, from 1 January 2016 to 30 October 2019.

#### 1.4.1. Database Debugging

The database collected some clinical data from each patient during 6-h hemodialysis sessions, which are labeled as “Hour 0”, “Hour 1”, “Hour 2”, “Hour 3”, “Hour 4”, and “Hour 5”. The way these data were collected and stored conformed to the original database, in the format shown on the left side of Figure 1. In this original format, the database consists of 656,370 rows and 28 columns (18,378,360 data). Each row contains, in its first six columns, some data identifying the patient and the session: identifier, sex, and age of the patient; whether the patient is hypertensive or diabetic; and date of the session. The remaining 22 columns contain the sequence of the different clinical parameters that are measured at the corresponding hour of the session. Thus, each session is composed of six consecutive rows.

However, to facilitate data processing by specialized software, we found it convenient to adapt the original format of the database to another format that shows, in each row, the entire session, so that the clinical information of each hour appears consecutively in the same row, as shown in the upper right side of Figure 1. An algorithm was programmed for this purpose. In addition, other additional algorithms were later applied to debug the database by removing duplicate sessions or sessions containing erroneous data.

Finally, the last algorithm calculated, for each session, the supposed hypotension suffered by the patient, from the systolic and diastolic blood pressures measured in each of the hours of the session. The label ‘YES’ or ‘NO’ of the possible hypotension in each hour of the session was included in the database, as shown in the bottom side of Figure 1.

Figure 2 shows some basic statistics of the debugged database. There are 758 patients, 69% male and 31% female. These patients correspond to 98,015 hemodialysis sessions, taking into account that not all patients record the same number of sessions.

Our main interest is to detect episodes of hypertension in these sessions. After calculating hypotension from systolic and diastolic blood pressures, we counted 25,026 sessions (26% of the total) where there was a hypotension episode in some of the hours of the session. These sessions correspond to 584 patients (77% of the total), of whom 25% are female and 52% male.

#### 1.4.2. Hypotension Calculus

The possible hypotension that occurs during a hemodialysis session is calculated from the systolic blood pressures measured in each of the 5 h of the session. If any of these pressures measured in “Hour 2”, “Hour 3”, and “Hour 4” is lower, by at least 20 mL, than any of the pressures measured in “Hour 0” and “Hour 1”, then there is hypotension, quantified by the Hypotension Measure (HYM) as the difference of the systolic blood pressures; otherwise, there is not. For example, if from “Hour 1” to “Hour 2” the systolic pressure has decreased by 25 mL, we say that there is a hypotension; on the other hand, if it has decreased by 18 mL, it is not considered hypotension.

Once we calculated the possible hypotension values in all the sessions, we could see two interesting results. Firstly, Figure 3 shows, just for each patient who had a hypotension episode (584), the percentage of sessions where he/she suffered this episode with respect to the total number of sessions he/she was subjected to. For example, a patient who received 294 sessions, in 34 of which a hypotension happened, has a rate of 12%. In this graph, the patients are arranged according to the percentage obtained. Thus, for example, we observe that most patients (75%) with hypotension showed it in fewer than 40% of their sessions or that almost 10% of these patients showed hypotension in all the sessions they were submitted to. Secondly, another interesting result relates to the possible influence of age on the measurement of hypotension. Figure 4 shows the average systolic blood pressure drop (HYM) in the sessions where patients experienced hypotension, arranged according to the age of the patients. The different calculated trend plots (linear and polynomial) always show a slight rise in the measurement of hypotension as age increases.

#### 1.4.3. Clinical Parameters

The database provides 28 parameters for each hemodialysis session. The first one corresponds to the patient’s identifier, conveniently anonymized, and the second one records the date of the session. Two parameters record whether the patient is hypertensive or diabetic; since no patient in the database suffers from either of these two pathologies, these parameters have not been considered when designing the predictive hypotension models.

The remaining 22 parameters are those needed to build the datasets for predictive analysis. The first two (sex and age) are patient identity parameters, while the remaining 20 are clinical parameters. Table 1 shows the list of these 22 parameters, with their corresponding initials and measurement units.

Finally, Figure 5 shows a set of graphics showing the value ranges of the clinical parameters. For a correct analysis of these graphs, it should be noted that, for each parameter, the sessions where it was measured are ordered according to the increasing value of the parameter.

## 2. Results

### 2.1. Generating Predictive Models from Machine Learning Classifiers

Machine Learning (ML) allows developing a wide set of algorithms for intelligent data analysis for predicting, among other purposes [25]. Its technology is particularly suited for analyzing medical data and making decisions in medical diagnostic problems [26]. For example, ML classifiers distinguish between healthy and Parkinson’s patients [27] for clinical diagnosis. The input of a ML process is a set of instances or datasets, represented as a matrix of instances versus attributes [28], and the output is the prediction result.

ML encompasses many learning methods sorted in two main groups: unsupervised learning and supervised learning [29]. In supervised learning, a labeled set of training data is used with to estimate or map the desired output. Particularly, our research falls into this group as a classification task, where the learning process categorizes data into a set of finite classes. Based on this process, each new sample can be categorized into one of the existing classes. In this study, we built predictive models from the clinical parameters as attributes or input variables, where the value of the predicted hypotension in a hemodialysis session (labeled as ’YES’ or ’NO’) is the categorical response or output.

We propose to generate predictive models from ML classifiers by using two different techniques: Decision Trees (DT) and Support Vector Machines (SVM).

#### 2.1.1. Decision Trees

Decision trees [30] are predictive representations that can be used for classification models. They are a hierarchical way of partitioning the space, where the goal is to create a model that predicts the value of a target variable based on several input variables. A DT learns by splitting the source set into subsets that are based on an attribute value test. This process is repeated on each derived subset in a recursive way, called recursive partitioning. When a DT is used for classification tasks, it is more appropriately referred to as a classification tree. This method has been successfully applied in many medical problems [31].

In this study, we built a fitted binary classification DT where the hyperparameters of the tree were automatically optimized. In addition, we considered standard Classification And Regression Tree (CART) algorithm to select the best split predictor at each node, which maximizes the split-criterion gain over all possible splits of all predictors. We note that trees grown using standard CART are not sensitive to predictor variable interactions.

#### 2.1.2. Support Vector Machines

Support Vector Machines [32] are supervised learning algorithms applied for classification tasks. A SVM classifies data by finding the best hyperplane that separates all data points according to two different classes. This optimal hyperplane has the largest margin between the two classes, where margin means the maximal width of the slab parallel to the hyperplane that has no interior data points.

We generate a SVM model for classification that supports sequential minimal optimization (SMO) [33], iterative single data algorithm (ISDA) [34], or L1 soft-margin minimization. It maps the predictor data by using kernel functions. Once the SVM is built and trained, we predict the hypotension according to the provided values of the clinical parameters.

### 2.2. Datasets

On the basis of the debugged database, it is necessary to build a dataset in the appropriate format to apply the supervised learning classifier. Basically, this dataset consists of a matrix where the rows are the sessions and the columns are the different clinical parameters (attributes), except for hypotension, which is the parameter to be classified and predicted (categorical response).

It is important to point out that, to build this dataset, only the sessions where all the values of the clinical parameters are valid have been taken into account, that is, parameters that have been correctly measured (they do not contain null values or values outside their own ranges). Not all sessions meet this condition; in fact, only between 12% and 28% of the sessions do, depending on the number of clinical parameters considered. The purpose of eliminating sessions with invalid data is to use datasets that generate the most robust prediction models possible. Obviously, this strategy will reduce the number of sessions available for the dataset, as shown in Figure 6.

This figure shows 12 experimental scenarios, each of which consists of an increasing number of parameters to build the dataset (from only 2 parameters in the first experiment to 24 parameters in the last one). The number of valid sessions decreases as more parameters are added to the dataset. Besides, this decrease also affects the number of valid sessions that are labeled with a positive hypertension, which also decreases. It should be noted that, for the correct design of a robust prediction model, it is not only necessary to have many valid sessions, but also a good number of valid sessions where hypertension occurred, since this is the parameter to classify and predict.

Table 2 shows the parameters considered for each of the 12 experimental scenarios, which correspond to a given dataset and we also call experiments. The identity parameters (sex and age) are present in all experiments, whereas the clinical parameters are included successively in the experiments, mostly in pairs, following a criterion of interest according to the possible influence of the clinical variables on the appearance of hypotension.

From the results in Figure 6 and Table 2, we can conclude that, although Dataset 12 contains all the clinical parameters, the size of the dataset is small enough to obtain a not very robust prediction model. Therefore, we discarded this experimental scenario for the application of the classifier and subsequent prediction analysis.

### 2.3. Experimental Procedure

The experimental procedure we designed to predict the appearance of hypotension during a hemodialysis session is outlined in Figure 7. Starting from the debugged database, as explained in Figure 1, we built up to 11 datasets, corresponding to the different number of selected clinical parameters (we excluded Dataset 12, as explained above).

Given any dataset, a specific part of it will be used by the classifier to build the prediction model (training dataset), while the remainder will be used to check the effectiveness of the model (test dataset). To obtain robust conclusions, we considered different percentages of the dataset dedicated to the training, while the remaining dataset is dedicated to the test. Therefore, we applied six experimental cases with different percentages for the training dataset: 70%, 75%, 80%, 85%, 90%, and 95%. These different percentages obviously have an impact on the number of sessions devoted to training and testing, as shown in Figure 8. Although we observed that 95% allows more sessions to be devoted to training, the fewer sessions available to validate the prediction could yield misleading results, hence the reason for examining these six scenarios.

As shown in Figure 7, the training dataset is submitted to the classifier to build the prediction model. Later, we apply the test sessions to the prediction model, obtaining the predicted hypotension. Therefore, we can compare these predictions with the real hypotensions of the test sessions, calculating the success percentage, which we use as the accuracy metric for the prediction.

Figure 9 and Figure 10 show the results of the prediction accuracy by success percentages, for each dataset and for each percentage of training dataset, for DT and SVM classifiers, respectively. It should be noted that, for each chosen percentage for the training dataset, the corresponding test sessions were selected randomly, whereas the training sessions were the remainder. This stochastic element of the experimentation requires a statistical analysis of the results, by repeating several times the prediction for each dataset and percentage of the training dataset. To this end, we performed 31 runs of each prediction. The figure shows the minimum, maximum, mean, median, standard deviation, and outliers for each experiment.

Table 3 summarizes some quality metrics for the classification, which help us to better analyze the prediction results. Having in mind that the positive and negative classes are the appearance or not of hypotension, respectively, accuracy measures the percent of cases in which the prediction was correct, whereas precision measures the quality of the classification model. Sensitivity shows how well the model can classify samples that have the condition, whereas specificity measures how well the model can identify true negatives. Following the above considerations, Figure 11 shows the mean values of the 31 runs for the quality metrics in the classification, according to the DT model and training size chosen.

## 3. Discussion

From the analysis of Figure 9 and Figure 10, we can draw interesting conclusions. Firstly, the prediction results are generally acceptable, in accuracy terms, where DT reaches slightly better results than SVM. Thus, we obtained mean accuracy rates ranging from 75% to 81% after applying DT, whereas these rates ranged from 74% to 80% when we used SVM. Secondly, we observed that the results of the last four datasets (“Datasets 8–11”) are very good, exceeding 80% of accuracy in all cases. This may be due to the robustness of their predictive models, since these datasets include more clinical parameters than the remaining seven datasets. From these four best datasets, we would choose “Dataset 10” as representative of the results of our research, since it has fewer outliers, less standard deviation, and a high number of clinical parameters considered. Finally, we found that the best results were obtained if we considered training datasets composed of 75% of the clinical dataset.

Besides accuracy, other quality metrics give us interesting information, as shown in Figure 11. Thus, precision values are always higher than 50% (reaching 60%), whereas specificity values are higher than 90% in all the cases. Sensitivity values are low, due to the low number of real hypotensions in the datasets with regard to negative cases.

These results encourage us to consider classifiers based on decision trees for predicting the appearance of hypotension in hemodialysis sessions. It should be considered that these predictive models were generated from much fewer sessions than those available in the database (around 25%), by considering only those sessions with all valid measures. This means that, if we include sessions with some unknown values for the parameters, we will be able to build other predictive models that complement the results obtained in our proposal.

As a limitation of our study, we consider that it is not advisable to use all the clinical parameters in a unique dataset to train the classifier, if we assume the constraint that all the values of all the clinical parameters in the training dataset must be known and valid, because then this dataset would be reduced to a very low number of sessions. As an alternative, we propose the generation of a collection of datasets, each of which considers different combinations of clinical parameters. In this way, all the clinical parameters are covered through these training datasets, each of which has a high number of sessions and, therefore, of cases where hypotension occurs, which improves the accuracy of the prediction.

Cardiovascular events can occur during a hemodialysis session, being responsible for about 50% of deaths of patients. Hypertension is easily detected and managed by available drugs, whereas hypotension is more difficult to prevent. This is the reason the prediction model proposed in this study can be an additional tool for helping healthcare personnel to prevent cardiovascular events during a hemodialysis session, being this approach the main contribution of our research.

## Figures and Tables

**Figure 1 ijerph-18-02364-f001:**
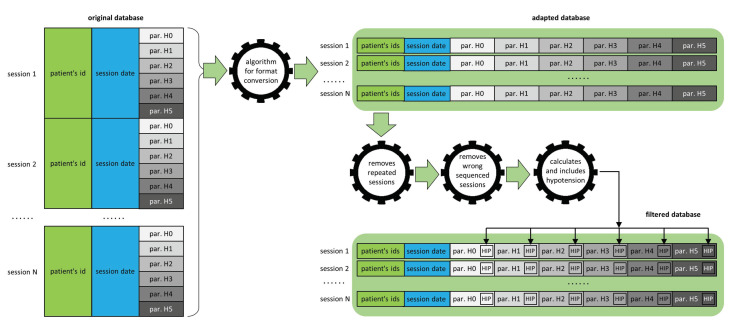
Algorithms for adapting and filtering the hemodialysis database for further processing.

**Figure 2 ijerph-18-02364-f002:**
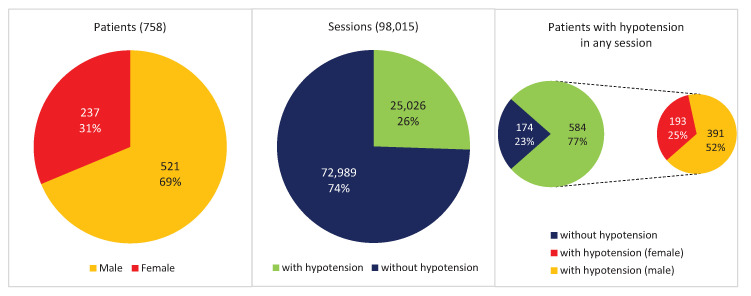
The main stats of the hemodialysis database.

**Figure 3 ijerph-18-02364-f003:**
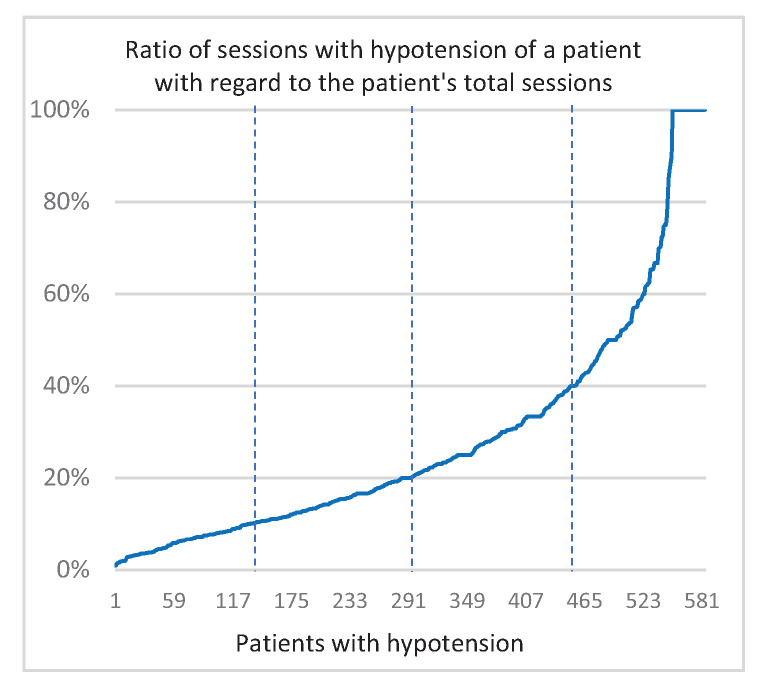
Ratio of patients’ sessions where hypotension appeared.

**Figure 4 ijerph-18-02364-f004:**
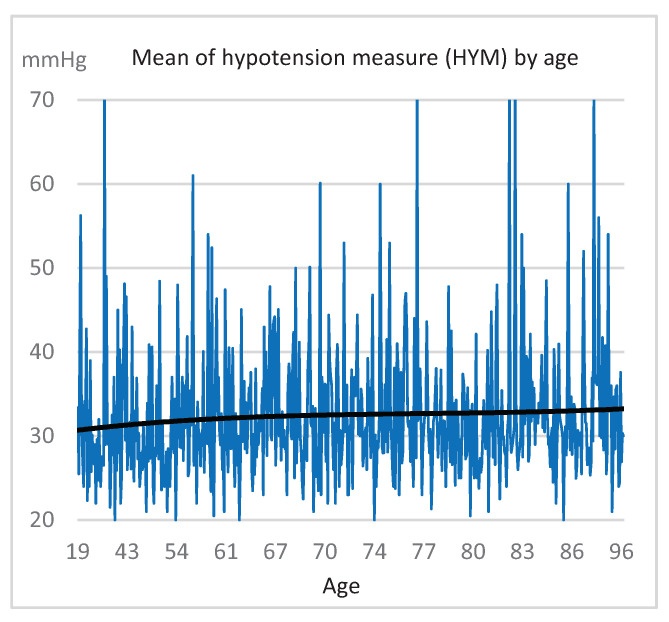
Average systolic blood pressure drop by age, and trend line.

**Figure 5 ijerph-18-02364-f005:**
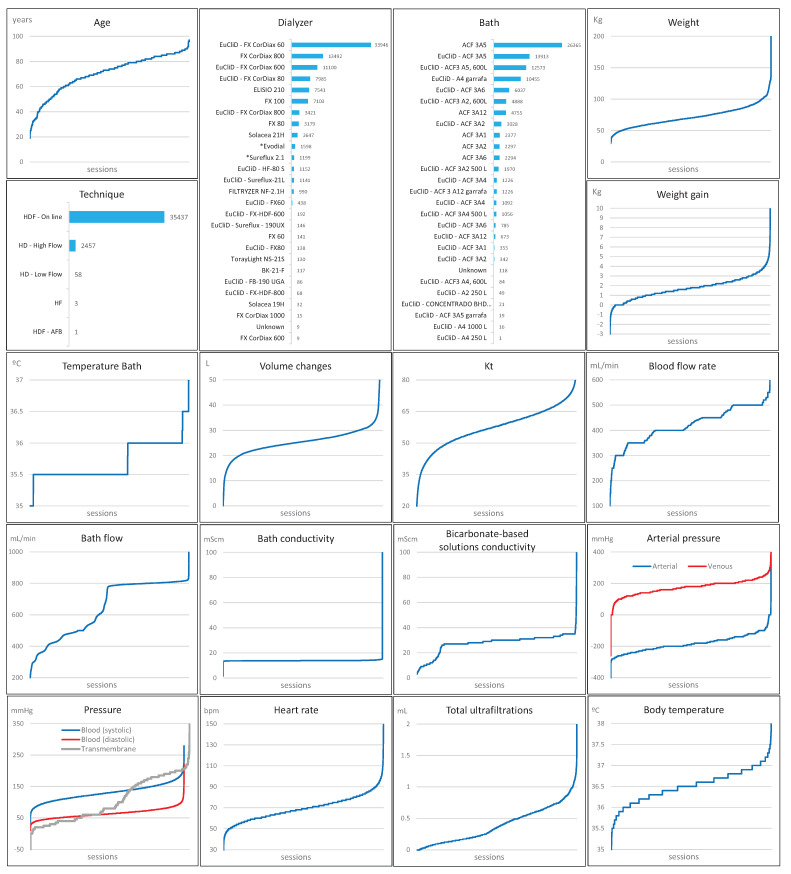
Ranges of the clinical parameters. Sessions arranged by increasing parameter values.

**Figure 6 ijerph-18-02364-f006:**
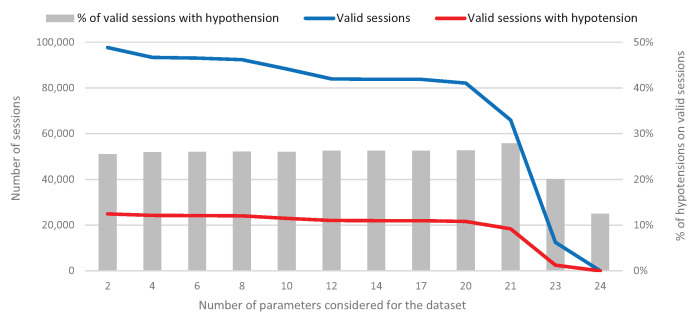
Datasets for different numbers of parameters considered. The number of valid sessions decreases when the number of parameters considered increases.

**Figure 7 ijerph-18-02364-f007:**
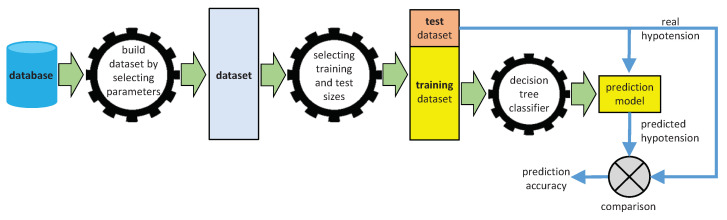
Prediction processing chain.

**Figure 8 ijerph-18-02364-f008:**
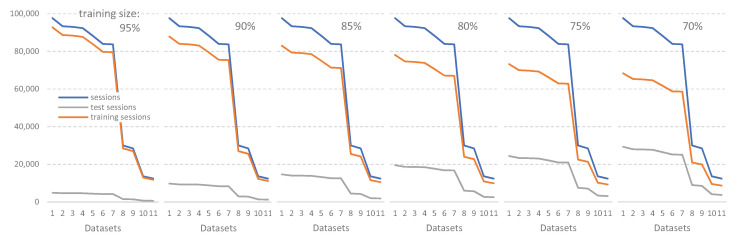
Sessions for training and test datasets, according to the selected training percentage.

**Figure 9 ijerph-18-02364-f009:**
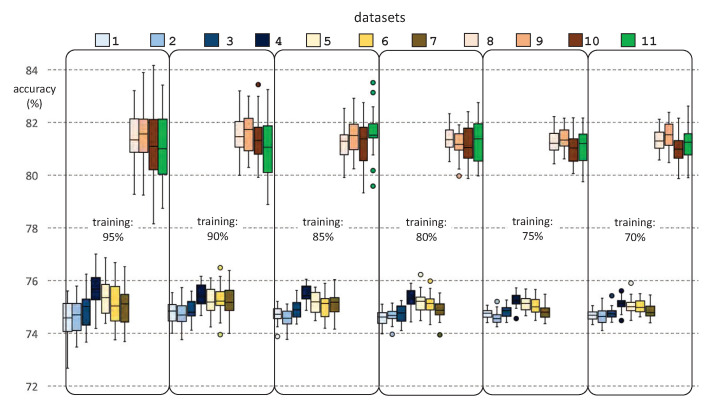
Prediction accuracy for each experiment and size of the training dataset, obtained from a decision tree classifier.

**Figure 10 ijerph-18-02364-f010:**
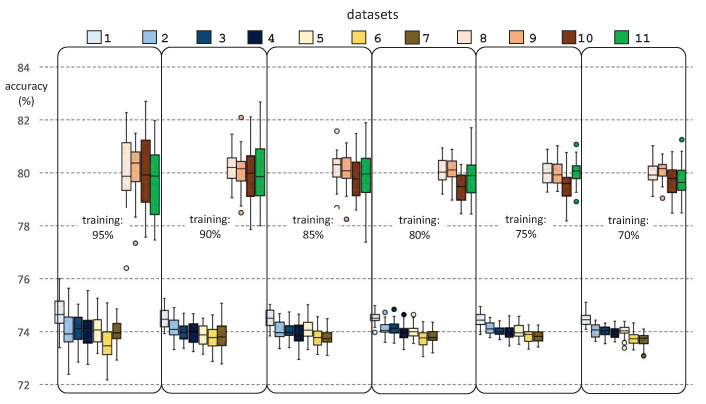
Prediction accuracy for each experiment and size of the training dataset, obtained from a support vector machine classifier.

**Figure 11 ijerph-18-02364-f011:**
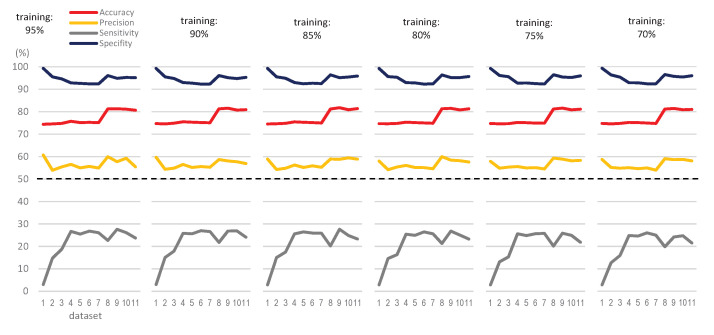
Quality metrics of the classification, according to the DT model and training size chosen.

**Table 1 ijerph-18-02364-t001:** Identity and clinical parameters.

Initials	Parameter	Units	Initials	Parameter	Units
SEX	Sex	‘Male’ or ‘Female’	BFR	Blood flow rate	mL/min
AGE	Age	years	HBF	Hemodialysis bath flow	mL/min
DIA	Dialyzer	dimensionless	HBC	Hemodialysis bath conductivity	mScm
BAT	Bath	dimensionless	BSC	Bicarbonate-based solutions conductivity	mScm
TEC	Type of technique	dimensionless	APR	Arterial pressure	mmHg
WDR	Weight (dry)	Kg	VPR	Venous pressure	mmHg
WPR	Weight (pre)	Kg	TMP	Transmembrane pressure	mmHg
WPO	Weight (post)	Kg	SBP	Systolic Blood Pressure	mmHg
IWG	Interdialytic weight gain	Kg	DBP	Diastolic Blood Pressure	mmHg
HBT	Hemodialysis bath temperature	°C	HRA	Heart rate	bpm
VOL	Volume changes	L	TUF	Total ultrafiltrations	mL
KT	Urea clearance	L	BOT	Body temperature	°C

**Table 2 ijerph-18-02364-t002:** Valid sessions (all the parameters are known and valid) for different number of parameters considered for building 12 different datasets.

Experiment/Dataset	1	2	3	4	5	6	7	8	9	10	11	12
Valid Sessions	97,640	93,355	93,013	92,351	88,265	83,949	83,745	83,745	82,097	65,938	12,469	32
Valid Sessions with Hypotension	24,912	24,242	24,168	24,050	22,953	22,025	21,969	21,969	21,612	18,397	2504	4
% of These Sessions	25.5%	26.0%	26.0%	26.0%	26.0%	26.2%	26.2%	26.2%	26.3%	27.9%	20.1%	12.5%
Parameters	2	4	6	8	10	12	14	17	20	21	23	24
SEX	x	x	x	x	x	x	x	x	x	x	x	x
AGE	x	x	x	x	x	x	x	x	x	x	x	x
DIA								x	x	x	x	x
BAT								x	x	x	x	x
TEC								x	x	x	x	x
WDR									x	x	x	x
WPR									x	x	x	x
WPO									x	x	x	x
IWG							x	x	x	x	x	x
HBT										x	x	x
VOL						x	x	x	x	x	x	x
KTV					x	x	x	x	x	x	x	x
BFR					x	x	x	x	x	x	x	x
HBF											x	x
HBC						x	x	x	x	x	x	x
BSC												x
APR			x	x	x	x	x	x	x	x	x	x
VPR			x	x	x	x	x	x	x	x	x	x
TMP							x	x	x	x	x	x
SBP		x	x	x	x	x	x	x	x	x	x	x
DBP		x	x	x	x	x	x	x	x	x	x	x
HRA				x	x	x	x	x	x	x	x	x
TUF				x	x	x	x	x	x	x	x	x
BOT											x	x

**Table 3 ijerph-18-02364-t003:** Quality metrics for the classification model.

Parameter		Meaning
True positive	TP	Hypotension predicted, which is correct
False positive	FP	Hypotension predicted, which is not correct
False negative	FN	No-hypotension predicted, which is not correct
True negative	TN	No-hypotension predicted, which is correct
Accuracy	$ACC=\frac{TP+TN}{TP+TN+FP+FN}$	Rate of correct predictions over total
Precision	$PRE=\frac{TP}{TP+FP}$	Rate of positive identifications actually correct
Sensitivity	$REC=\frac{TP}{TP+FN}$	Rate of positives correctly predicted
Specificity	$SPE=\frac{TN}{TN+FP}$	Rate of negatives misclassified

## Data Availability

Data supporting reported results can be found in: Gomez-Pulido, Juan A. (2021), “Dialysis database: sessions with valid data of clinical parameters”, Mendeley Data, V1, doi: 10.17632/7kmtsmsgfw.1.

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
