# Peer review of "Predicting the Appearance of Hypotension during Hemodialysis Sessions Using Machine Learning Classifiers"

_ijerph, 2021, doi:10.3390/ijerph18052364_

Round 1

Reviewer 1 Report

The paper presents some predictive models to predict the appearance of hypotension during a dialysis session. The objectives of this study are clearly defined.

From a technical point of view the proposed method is standard and the technique is well known. The application of the technique in field of dialysis is interesting but the discussion is weak. The authors didn’t include specificity and sensitivity in the results. These concepts are important for evaluating the models' performance and should be presented. Moreover, which are the specific methods for decision trees? Are they C4.5 decision trees? They should be specified. Which tools have been used?

In general, the Section 1 is very repetitive and boring. It should be shortened and better explained. In the background section miss specific references to papers proposing the usage of ML methods in the field of dialysis for predicting hypotension.

Author Response

Dear reviewer: We are grateful for your work committed to the review of this paper. Next, we expose our amendments and/or explanations to all the issues you have pointed out; they have encouraged us to improve the manuscript.

Comment: The paper presents some predictive models to predict the appearance of hypotension during a dialysis session. The objectives of this study are clearly defined. From a technical point of view the proposed method is standard and the technique is well known. The application of the technique in field of dialysis is interesting but the discussion is weak.

Answer: Thank you for your comments. We have improved the discussion and results. Now, we include new results after applying a new classifier, based on support vector machines. Also, the discussion was extended with some considerations about the clinical application, limitations of our study, etc.

Comment: The authors didn’t include specificity and sensitivity in the results. These concepts are important for evaluating the models' performance and should be presented. Moreover, which are the specific methods for decision trees? Are they C4.5 decision trees? They should be specified. Which tools have been used?

Answer: We generated a fitted binary classification decision tree based on the attributes (the clinical parameters) as input variables, and the categorical response (the value of hypotension, 'YES' or 'NO') as output. The hyperparameters of the tree were automatically optimized. We considered standard Classification And Regression Tree (CART) algorithm to select the best split predictor at each node, which maximizes the split-criterion gain over all possible splits of all predictors. We note that trees grown using standard CART are not sensitive to predictor variable interactions.

Comment: In general, the Section 1 is very repetitive and boring. It should be shortened and better explained.

Answer: We have shortened following your suggestion. We summarized the subsection about experiences of massive data analysis and pattern recognition in healthcare, and removed the references to several projects related to Big Data developed in the field of digital health in Europe, since they do not provide substantial information regarding the research we have carried out.

Comment: In the background section miss specific references to papers proposing the usage of ML methods in the field of dialysis for predicting hypotension.

Answer: We have referenced some studies regarding to the presence of hypotension during the hemodialysis sessions, although they do not predict specifically the hypotension from machine learning classifiers, which is the focus of our research: [8][19][20][21][22][23][24].

Reviewer 2 Report

The authors collected 758 patients' clinical information and constructed a model to predict hypotension by 22 clinical parameters. This work is meaningful and has clinical significance. However, I suggest the authors to further consider the following issues:

1) For the prediction model, 70% of the data set is generally used as the training set and 30% as the test set. It is suggested to show the results of the prediction model in this case.

2) There are many ML methods to be used. It is suggested that the author use other ML algorithms, such as linear model, K-NN and support vector machines, to build the prediction models and demonstrate their results.

Author Response

Dear reviewer: We are grateful for your work committed to the review of this paper. Next, we expose our amendments and/or explanations to all the issues you have pointed out; they have encouraged us to improve the manuscript.

Comment: The authors collected 758 patients' clinical information and constructed a model to predict hypotension by 22 clinical parameters. This work is meaningful and has clinical significance. However, I suggest the authors to further consider the following issues: 1) For the prediction model, 70% of the data set is generally used as the training set and 30% as the test set. It is suggested to show the results of the prediction model in this case.

Answer: Thank you for your comments. We have repeated the experiments by including two new cases, where training dataset are composed of 70% and 75% of the clinical datataset. In the discussion section we comment that the best results were obtained considering 75%, almost the same than for 70%.

Comment: 2) There are many ML methods to be used. It is suggested that the author use other ML algorithms, such as linear model, K-NN and support vector machines, to build the prediction models and demonstrate their results.

Answer: Following your suggestion, we repeated all the prediction experiments by applying support vector machines. We included new figures and results in the manuscript. We discuss that DT reaches slightly better results than SVM.

Reviewer 3 Report

This paper talks about predicting the appearance of hypotension during hemodialysis sessions using decision tree classifiers. This work presents an investigation to predict the appearance of hypotension during a dialysis session, using predictive models trained from a large dialysis database, which contains the clinical information of 98,015 sessions corresponding to 758 patients. The prediction model takes into account up to 22 clinical parameters measured 5 times during the session, as well as the gender and age of the patient. I read the manuscript with great interest and believe its topic is important and relevant. The authors performed a careful and thorough review of the literature, as the section was very informative and substantial. Appropriate theoretical framework was applied. I found the methodological part to be well justified and reasonable for this type of analysis. Although the manuscript is overall well-written and structured, it might benefit from additional spell/language checking. However, I have some comments which I would like to be addressed before the acceptance of this paper.

  • The merit of the proposed approach is supported by the results, but I miss on the paper a bit more discussion on why these techniques were chosen for this problem and had not been considered before. This however is more of a nitpicking than a detrimental comment.
  • Another concern is that, I fail to see any reference to the availability of the models, the data or the source code. Without this information it is impossible to independently verify or reproduce any of your claims, and the article greatly suffers in its utility and credibility.
  • What was the key motivation behind choosing the Decision Tree Classifiers?
  • On page 5 authors mentioned about “In recent years several projects related to Big Data have been developed in the field of digital health in Europe……”. If authors can report the findings of the above mentioned projects that will be great for readers otherwise, this information is not worthy.
  • It would be interesting if the authors report the trade-off compared to other methods especially the computational complexity of the models. Some techniques require more memory space and take longer time, please elaborate on that.
  • Authors should further clarify and elaborate novelty in their contribution.
  • Besides do compare the present study findings with the past studies that used same datasets and highlight why your work is better. And in this context it is worth mentioning their experimental evaluation protocol for a fair comparison.
  • What are the limitations of the preset study?
  • What are the practical implications and how medical paractioners get benefit from present study findings? Add them in the discussion section?
  • There are several interesting papers that look into healthcare. For instance, the below papers has some interesting implications that you could discuss in your Introduction and how it relates to your work.
  • Ali, Farman, et al. "A smart healthcare monitoring system for heart disease prediction based on ensemble deep learning and feature fusion." Information Fusion 63 (2020): 208-222.
  • Alsinglawi, Belal, et al. "Predicting Length of Stay for Cardiovascular Hospitalizations in the Intensive Care Unit: Machine Learning Approach." 2020 42nd Annual International Conference of the IEEE Engineering in Medicine & Biology Society (EMBC). IEEE, 2020.
  • Ijaz, Muhammad Fazal, Muhammad Attique, and Youngdoo Son. "Data-Driven Cervical Cancer Prediction Model with Outlier Detection and Over-Sampling Methods." Sensors10 (2020): 2809.

Author Response

Dear reviewer: We are grateful for your work committed to the review of this paper. Next, we expose our amendments and/or explanations to all the issues you have pointed out; they have encouraged us to improve the manuscript.

Comment: This paper talks about predicting the appearance of hypotension during hemodialysis sessions using decision tree classifiers. This work presents an investigation to predict the appearance of hypotension during a dialysis session, using predictive models trained from a large dialysis database, which contains the clinical information of 98,015 sessions corresponding to 758 patients. The prediction model takes into account up to 22 clinical parameters measured 5 times during the session, as well as the gender and age of the patient. I read the manuscript with great interest and believe its topic is important and relevant. The authors performed a careful and thorough review of the literature, as the section was very informative and substantial. Appropriate theoretical framework was applied. I found the methodological part to be well justified and reasonable for this type of analysis. Although the manuscript is overall well-written and structured, it might benefit from additional spell/language checking. However, I have some comments which I would like to be addressed before the acceptance of this paper.

Answer: Thank you very much for your valuable comments. The updated version of the manuscript includes new experiments and results, summarizes some sections, extends the discussion, clarifies some paragraphs, etc.

Comment: The merit of the proposed approach is supported by the results, but I miss on the paper a bit more discussion on why these techniques were chosen for this problem and had not been considered before. This however is more of a nitpicking than a detrimental comment.

Answer: Section "Techniques massive data analysis and its applicability to the field of hemodialysis" discusses some considerations about the use of data mining and machine learning techniques to this end. However, our approach deals with ML classifiers due to the particular characteristics of the clinical database considered, and the goal of predicting the appearance of the hypotension according to the evolution of the clinical parameters along the different hours of a hemodialysis session, which is an approach not considered before.

Comment: Another concern is that, I fail to see any reference to the availability of the models, the data or the source code. Without this information it is impossible to independently verify or reproduce any of your claims, and the article greatly suffers in its utility and credibility.

Answer: Thank you for your comment, we agree that it is necessary to share the information with the scientific community. To this end, we have submitted the datasets, along with the corresponding instructions, to Mendeley Data Repository. It can be accessed and referenced as follows: http://dx.doi.org/10.17632/7kmtsmsgfw.1 // Gomez-Pulido, Juan A. (2021), “Dialysis database: sessions with valid data of clinical parameters”, Mendeley Data, V1, doi: 10.17632/7kmtsmsgfw.1

Comment: What was the key motivation behind choosing the Decision Tree Classifiers?

Answer: At the beginning, we chose DT because it is a robust classifier for predicting purposes when large datasets are considered. However, for this revised manuscript, we have included new prediction experiments by applying support vector machines, pursiung more reliable results. We have included new figures and results in the manuscript, and we discuss that DT reaches slightly better results than SVM.

Comment: On page 5 authors mentioned about “In recent years several projects related to Big Data have been developed in the field of digital health in Europe……”. If authors can report the findings of the above mentioned projects that will be great for readers otherwise, this information is not worthy.

Answer: You are right. We have removed the references to these projects, also for summarizing the section.

Comment: It would be interesting if the authors report the trade-off compared to other methods especially the computational complexity of the models. Some techniques require more memory space and take longer time, please elaborate on that.

Answer: Thank you for your suggestion. This is another reason to consider an additional classifier different than DT. We chose support vector machines because it requires different computing workload than DT and allow a fair comparison with the previous results. As we pointed out before, the prediction results of DT and SVM were very similar, although DT reached slightly better results than SVM.

Comment: Authors should further clarify and elaborate novelty in their contribution.

Answer: We have added a comment in the Discussion section to highlight that the main contribution of our research is to provide a useful tool for helping healthcare personnel to prevent cardiovascular events during a hemodialysis session.

Comment: Besides do compare the present study findings with the past studies that used same datasets and highlight why your work is better. And in this context it is worth mentioning their experimental evaluation protocol for a fair comparison.

Answer: This study cannot yet be compared with other works, because it is based on a large novel clinical database composed of hemodialysis sessions that collect the same parameters along different hours. We have not found a similar approach, in addition to the fact that this database has not yet been used by any other scientific study.

Comment: What are the limitations of the preset study?

Answer: We have highlighted at the end of the Discussion section, as a limitation of our study, that it is not advisable to use all the clinical parameters in a unique dataset to train the classifier, if we assume the constraint that all the values of all the clinical parameters in the training dataset must be known and valid, because then this dataset would be reduced to a very low number of sessions. As an alternative, we have proposed the generation of a collection of datasets, each of which considers different combinations of clinical parameters. In this way, all the clinical parameters are considered through these training datasets, each of which has a high number of sessions and, therefore, of cases where hypotension occurs, which improves the accuracy of the prediction.

Comment: What are the practical implications and how medical paractioners get benefit from present study findings? Add them in the discussion section?

Answer: We have included the following paragraph in the discussion section: "Cardiovascular events can occur during a hemodialysis session, being responsible for about 50% of deaths of patients. Hypertension is easily detected and managed by available drugs, whereas hypotension is more difficult to prevent. This is the reason why the prediction model proposed in this study can be an additional tool for helping healthcare personnel to prevent cardiovascular events during a hemodialysis session, being this approach the main contribution of our research".

Comment: There are several interesting papers that look into healthcare. For instance, the below papers has some interesting implications that you could discuss in your Introduction and how it relates to your work: Ali, Farman, et al. "A smart healthcare monitoring system for heart disease prediction based on ensemble deep learning and feature fusion." Information Fusion 63 (2020): 208-222. Alsinglawi, Belal, et al. "Predicting Length of Stay for Cardiovascular Hospitalizations in the Intensive Care Unit: Machine Learning Approach." 2020 42nd Annual International Conference of the IEEE Engineering in Medicine & Biology Society (EMBC). IEEE, 2020. Ijaz, Muhammad Fazal, Muhammad Attique, and Youngdoo Son. "Data-Driven Cervical Cancer Prediction Model with Outlier Detection and Over-Sampling Methods." Sensors10 (2020): 2809.

Answer: Thank you for this information. We have included the three references as practical examples in the section " Evidences supporting the use of massive data analysis in Healthcare".

Round 2

Reviewer 1 Report

The authors did not report the sensitivity and specificity values of the machine learning models.

Sensitivity and specificity are two measures used together to measure the predictive performance of a classification model. For example, sensitivity measures the percentage of instances of class "yes" (i.e., positive examples) correctly classified; and specificity measures the percentage of instance of class "no" (i.e., negative examples) misclassified. They are defined with reference to a confusion matrix.

In many problems, especially in clinical domains, classification accuracy is not enough to evaluate a model.

For example, in an unbalanced dataset, a model can classify correctly all the instances of the majority class and achieve a high classification accuracy, but misclassified the instances of the minority class.

Author Response

Thank you for your valuable comment. At the end of the section "Experimental Procedure", we have added a table that summarizes some quality metrics for the classification: accuracy, precision, sensitivity and specificity. They help us to better analyze the prediction results. Having in mind that the positive and negative classes are the appearance or not of hypotension respectively, we explain the meaning of these parameters. Furthermore, we added a new figure that shows the mean values of the 31 runs for these quality metrics in the classification, according to the DT model (which provided the best results) and training size chosen. Besides, in the section "Discussion" we discuss these results.

Reviewer 2 Report

all my comments are addressed.

Author Response

Thank you very much for your revision.

Reviewer 3 Report

All my comments are addressed hence, manuscript is accepted. 

Author Response

Thank you very much for your revision.